# Lipid Nanoparticles as Delivery Vehicles for Inhaled Therapeutics

**DOI:** 10.3390/biomedicines10092179

**Published:** 2022-09-02

**Authors:** Ellenmae W. X. Leong, Ruowen Ge

**Affiliations:** Department of Biological Sciences, Faculty of Science, National University of Singapore, Singapore 117558, Singapore

**Keywords:** lipid nanoparticles (LNPs), inhalation drug delivery, lung, respiratory diseases

## Abstract

Lipid nanoparticles (LNPs) have emerged as a powerful non-viral carrier for drug delivery. With the prevalence of respiratory diseases, particularly highlighted by the current COVID-19 pandemic, investigations into applying LNPs to deliver inhaled therapeutics directly to the lungs are underway. The progress in LNP development as well as the recent pre-clinical studies in three main classes of inhaled encapsulated drugs: small molecules, nucleic acids and proteins/peptides will be discussed. The advantages of the pulmonary drug delivery system such as reducing systemic toxicity and enabling higher local drug concentration in the lungs are evaluated together with the challenges and design considerations for improved formulations. This review provides a perspective on the future prospects of LNP-mediated delivery of inhaled therapeutics for respiratory diseases.

## 1. Introduction

The lungs form a crucial barrier between the body’s internal physiology and the outside world. As such, this interface is constantly under great stress, defending against infectious pathogens and foreign particulate matter. Respiratory diseases, both acute and chronic, are some of the most familiar and widespread human afflictions. Direct drug delivery via the airway has demonstrated advantageous efficacy for several diseases of the respiratory tract and lung. With the advancement of nanotechnology, lipid nanoparticles (LNPs) have emerged as a powerful non-viral carrier for drug delivery. In this review, we evaluate the current progress in using LNPs to deliver inhaled therapeutics and the challenges that remain. The technologies of LNP generation, which has been previously reviewed extensively, will not be our focus here.

### 1.1. Lung Diseases

The lung is directly exposed to the outside environment through the airways. It contains two main functional parts, the conducting zone (trachea, bronchi and bronchioles) and respiratory zone (alveoli). The top five most common lung diseases causing severe illness and death worldwide include tuberculosis, respiratory infections, lung cancer, asthma and chronic obstructive pulmonary disease (COPD), which together have a huge global burden [1]. Asthma is the most common respiratory affliction in children, affecting 262 million people around the world in 2019 and was the cause of 455,000 deaths [2]. Infections are the fourth leading cause of death whereas COPD, an umbrella term for multiple conditions leading to reduced airflow, is the third and responsible for three million deaths annually [3,4]. Less common but also severe chronic lung diseases include cystic fibrosis (CF), lymphangioleiomyomatosis (LAM) and pulmonary arterial hypertension (PAH). Over the past two years, the COVID-19 pandemic has brought to the forefront just how quickly and direly lung function can be impaired following an acute respiratory infection. The pandemic has also highlighted the bidirectional relationship between acute and chronic lung conditions [1,4,5]. For example, many recovered COVID-19 patients will have to contend with long-term respiratory side effects including pneumonia and acute respiratory distress syndrome (ARDS) [6,7]. The pathophysiology of ARDS includes lower airway inflammation, depleted lymphocytes, alveolar oedema, severe breathing difficulties and lasting pulmonary scarring [6,7,8,9]. Concurrently, pre-existing chronic lung diseases such as COPD increase susceptibility to a COVID-19 infection and the likelihood of severe disease [7]. Together with the 370 million COVID-19 cases and over 5.6 million deaths globally as of 30 January 2022, the pandemic has emphasized the importance of effective therapeutic treatments for lung diseases [10].

Bronchodilators are frequently employed to address pathologically narrowed airways [2]. Long-acting and short-acting β2-adrenergic agonists (LABAs and SABAs) activate β2-adrenergic receptors to induce relaxation in the airway smooth muscle tissue where these receptors are highly expressed [11,12]. The other major class of bronchodilators are long-acting and short-acting muscarinic antagonists (LAMAs and SAMAs) that block the activation of muscarinic acetylcholine receptors on airway smooth muscle cells, preventing their contraction [12,13]. In recent clinical applications, long-acting LABAs and LAMAs have been preferred [11,14]. Concurrently, novel corticosteroids such as fluticasone propionate in Advair^®^ (GlaxoSmithKline, Brentford, Middlesex, United Kingdom) and ciclesonide in Alvesco^®^ (Covis Pharma, Luxembourg) aid in reducing airway inflammation [2,14]. Although inhaled corticosteroids (ICS) are a mainstay asthma treatment, there has been conflicting evidence regarding their usefulness in COPD treatment due to the increased risk of bacterial and fungal respiratory infections [15,16,17]. The current evidence suggests that ICS use in combination with LABAs is beneficial in certain patient subgroups where pre-existing severe COPD exacerbations warrant more aggressive treatment [16,17,18,19]. Accurate patient history and assessment of COPD progression are instrumental to guide disease management. Patients are also frequently prescribed antibiotics to counter bacteria proliferation in the airway mucus [19]. Pulmonary delivery by single- or multiple-drug-loaded inhaler devices is the standard-of-care for asthma and COPD [18]. This increases local drug delivery while limiting systemic drug concentrations [20].

Despite the successful use of the pulmonary route for drug delivery, it remains a challenge to overcome the physiological barriers and robust clearance mechanisms in the respiratory system. The mucus gel layer and clearance by the beating of cilium in the airways limit lung retention times of naked drugs [14,21]. Larger drug doses or more frequent dosing regimens are thus necessary to achieve the desired therapeutic effect, which not only increases side effects but also reduces long-term patient adherence [22]. This has prompted the development of drug encapsulation technologies to increase drug stability and retention at the airway target sites [23]. 

### 1.2. Nanotechnology, Nanomedicine and Lipid Nanoparticles

The advent of nanotechnology enabled the engineering of materials and particles at the nanometre scale [24,25]. This has been pivotal for advancing drug delivery for the treatment of human diseases [25,26]. Drug development transformed as therapeutics were no longer limited by their solubility and stability within biological fluids for successful in vivo absorption and distribution [27]. Drug nanocarriers could be customised to provide protective and functional improvements to drug stability and pharmacokinetics [27]. Among the earliest and most promising drug nanocarriers are lipid nanoparticles (LNPs) [24,28]. LNPs are small, artificial, spherical assemblies of physiologically compatible lipids, making them the least toxic class of nanocarriers [29,30]. Consequently, they have been the subject of intense research in intravenous, intramuscular, oral, and pulmonary drug delivery [23,31,32,33].

LNPs comprise of therapeutic agents placed inside the lipid coatings typically made of phospholipids, cholesterol, polyethylene glycol (PEG)-conjugated lipids and ionisable cationic lipids with varying arrangements [34,35]. Ionisable cationic lipids provide the positive charge needed to electrostatically interact with negatively charged payloads and cellular membranes for uptake and payload release [27,34,35]. They possess a lower surface charge than constitutively charged cationic lipids and are neutral at physiological pH, which reduces toxicity, immune system activation and clearance [27,33,34,36].

The earliest class of LNPs developed were liposomes. Liposomes were introduced to clinical use in the 1980s as the bases for topical ointments, lotions and creams following their application in cosmetics [27,37]. The lipid components in a liposome are arranged such that a concentric lipid bilayer structure made of amphipathic phospholipids surrounds an aqueous interior, enabling both hydrophobic and hydrophilic therapeutic payloads to be embedded (Figure 1A,B) [28,30,35,36,38]. Dipalmitoylphosphatidylcholine (DPPC), distearoylphosphatidylcholine (DSPC) and distearoylphosphaethanolamine (DSPE) are phospholipids commonly used in synthetic liposomes that form the bilayer structures with cholesterol [28,30,39,40].

LNPs were further explored with the reduction in their size and increase in the variety of structures, chemistries and applications [27]. For example, liposomes have since spurred the development of two other categories of LNPs: solid lipid nanoparticles (SLNs) and nanostructured lipid carriers (NLCs) [35]. SLNs and NLCs feature a lipid shell surrounding a hydrophobic core instead of an aqueous one, able to carry hydrophobic therapeutics (Figure 1E,F) [30,41]. SLNs possess solid hydrophobic cores that can contain additional components such as glycerides, esters and waxes (Figure 1E) [28,30,41]. NLCs have been called the second generation SLN [36]. The hydrophobic cores of NLCs are of an imperfect crystal or amorphous structure, containing a mix of solid and liquid lipids that reduces the mobility of incorporated drugs, thereby enhancing drug loading and controlled drug release capacities (Figure 1F) [28,30,36]. LNPs typically have an aerodynamic diameter between 100 and 300 nm but smaller (<100 nm) or larger particles (up to 1000 nm) can also be obtained [23,28,36,41]. For example, SLNs can range from 50 nm to 1000 nm, whereas NLCs typically have diameters between 100 and 500 nm [30]. Liposome diameters can range from 250 nm all the way up to 2.5 μm [29]. The structural diversity of LNPs has fuelled the possibilities for their therapeutic applications. This review will thus assess the potential of using LNPs as a pulmonary drug delivery method.

## 2. History and Development of LNP Therapeutics

The first LNP drug approved by the U.S. Food and Drug Administration (FDA) was the liposome-encapsulated doxorubicin, Doxil^®^ (Johnson & Johnson, New Brunswick, NJ, USA) in 1995, for the treatment of ovarian cancer, metastatic breast cancer, AIDS-associated Kaposi’s sarcoma and multiple myeloma [30,36,38,42,43]. The intravenously administered liposomal formulation consists of hydrogenated soya phosphatidylcholine, cholesterol and PEG-modified DSPE [28,35,44,45]. Doxil^®^ achieved accumulation within tumours and had greater anti-tumour efficacy and reduced cardiotoxicity compared with free doxorubicin due to the enhanced permeability and retention (EPR) effect in tumours [35,36,42]. EPR is a phenomenon where molecules of certain sizes (typically liposomes, nanoparticles and macromolecular drugs) tend to accumulate in tumour tissue much more efficiently than they do in normal tissues due to the higher vascular permeability of tumour blood vessels [42,46]. Since then, the accumulative research on LNP lipid component diversity, targeted LNPs with ligand-modified surfaces (Figure 1C), PEG-conjugated immune-evasive LNPs (Figure 1D) and stimuli-responsive LNPs has enabled the development of effective LNP drug carrier systems for the treatment of more diseases [28].

Whereas next-generation SLN and NLC formulations constitute the fastest growing area of LNP therapeutic research owing to their aforementioned improved drug carrier characteristics, liposomes currently still form the majority of FDA-approved drugs following Doxil^®^ [28]. To name a few: liposomal amphotericin B for Aspergillosis infections (AmBisome^®^, Gilead Sciences, Foster City, CA, USA); liposomal verteporfin for macular degeneration (Visudyne^®^, Novartis AG, Basel, Switzerland); and liposomal bupivacaine as a non-opioid local nerve block (Exparel^®^, Pacira BioSciences, Tampa, FL, USA) [33,35,38,42]. Liposomes have also been applied in nucleic acid delivery where they first demonstrated their promise in siRNA-mediated liver gene silencing in non-human primates [33,47]. Subsequently, Alnylam Pharmaceuticals’ liposomal siRNA drug ONPATTRO^®^ (Cambridge, MA, United States) for hereditary transthyretin-mediated amyloidosis was approved by the FDA in 2018 [34,36]. Research was also conducted on non-hepatic targets of LNP-nucleic acid therapy such as neurons, immune cells, osteoclasts and osteoblasts [27,48,49,50,51,52]. Rungta et al. reported silencing of the GluN1 subunit of the neuronal N-methyl-D-aspartate receptor after delivering GluN1 siRNA in LNPs to mice via intracranial injection [48]. GAPDH and CD45 in macrophages and dendritic cells were also silenced by gene-specific siRNA delivered within LNPs via mouse tail vein injections [49]. With the ability to target immune cells, LNP-mRNA vaccines against cancer and infectious diseases were also being developed [52]. Moderna first tested mRNA vaccines against H10N8 and H7N9 influenza strains delivered in LNPs and found them to be well tolerated and immunogenic in randomized, placebo-controlled, double-blind, Phase 1 clinical trials (NCT03076385 and NCT03345043) [51]. These successes paved the way for Moderna and Pfizer–BioNTech gaining FDA-approval for their liposomal COVID-19 mRNA vaccines delivered via intramuscular injections [35,53].

As systemically administered LNP drugs progressed, so did inhaled LNPs targeting lung conditions. Evidence supporting the safety and efficacy of inhaled LNP-encapsulated drugs emerged in the 1990s [23,54,55,56,57,58,59,60,61]. Interest in them has been growing on account of the bio-similarity of lipid membranes. Lipid coatings aid nanoparticles for cellular uptake, penetrating pulmonary barriers and overcoming the clearance mechanisms, thereby boosting inhaled drug retention times in the airways [21,62]. Additional abilities such as protecting drugs from degradation and prolonging drug release time from the nanoparticles, make LNPs a forerunner delivery system for a wide variety of drugs [21,62]. LNPs have demonstrated the ability to carry small molecule drugs, nucleic acids and protein/peptide therapeutics into the deep lung tissue with good dispersion, retention time and cytotoxic activity against target cells [18,41,63,64,65]. Using LNP to deliver chemotherapy drugs to resident lung tumours, bronchodilators and mucolytics for the treatment of obstructive pulmonary diseases as well as antibiotics and gene therapy drugs have been the major areas of research in recent years [18,41,63,64,65,66,67]. The antibiotic amikacin showed greatly enhanced antibiotic exposure in the lungs when delivered in inhalable liposomes over intravenously administered free amikacin [68]. An inhaled liposomal amikacin formula has since been FDA-approved in 2018 to treat Mycobacterium avium complex infections (Arikayce^®^, Insmed Inc., Bridgewater, NJ, USA) [69,70]. It is a common nontuberculous mycobacterium infection following chronic lung conditions like COPD, asthma, CF and bronchitis [68,69,70]. The combination of nanomedicine and inhalation therapy thus offers much promise for improving chronic lung disease management and patient outcomes [23].

## 3. Developments in Inhaled LNP Therapeutics for Respiratory Diseases

### 3.1. Small Molecule Drugs

The majority of the currently administered drugs to manage chronic lung diseases like COPD and asthma are aerosolized small molecule drugs such as β2-adrenergic agonists, antimuscarinic agents and ICS [2,14]. However, robust lung clearance mechanisms limit the retention time and therapeutic window of naked drugs [14,21,22]. Encapsulation within LNPs can improve the local concentration and retention times of COPD and asthma drugs in the lungs [62]. Small molecule drugs tend to be hydrophobic and thus are poorly water soluble [71,72]. Anticancer drug candidates are frequent members of this category [64,72,73,74]. Encapsulating hydrophobic drugs within LNPs, particularly SLNs and NLCs, can help to improve drug solubility and bioavailability [30,39]. In this regard, studies in recent years have experimented with employing LNP encapsulation to improve the pulmonary delivery of anti-inflammatory agents, antibiotics and anti-cancer drugs to treat a variety of respiratory diseases.

The worldwide prevalence of air pollution and exposure to harmful gases pose a risk of exacerbating the symptoms of lung diseases [1]. If not managed, smoke-induced inflammation can lead to further deterioration of lung function [1]. In 2019, a group studied the effects of inhaled SLNs carrying the antioxidant carvacrol in mouse models of smoke-induced lung damage [75]. This damage model is characterized by the release of pro-inflammatory cytokines, damaged alveoli, shortness of breath and the presence of laryngeal exudate [75]. Mice that received the SLN drugs via nebulization delivery experienced improved local delivery and enhanced anti-inflammatory function of carvacrol, resulting in reduced pulmonary emphysema, exudate and oxidative damage in their lungs [75].

Many lung diseases involve chronic inflammation, which presents thick mucus that promotes bacterial proliferation and infections [19]. The prospect of delivering LNP-packaged antibiotics locally to the lungs was cemented with the approval of Arikayce^®^ (Insmed Inc.) to treat nontuberculous mycobacteria infections [68,69,70]. LNPs were also employed to deliver antibiotics targeting *Mycobacterium tuberculosis* residing within alveolar macrophages [66,76,77]. In a 2019 study, rats inhaled fluorescently-labelled nanoliposomes carrying the fluoroquinolone antibiotic moxifloxacin using a dry powder inhaler [76]. In vivo deep rat lung tissue deposition of this drug was confirmed, coupled with observed in vitro anti-tubercular activity and in vitro murine alveolar macrophage drug uptake [76]. Moreover, variations to the nanoliposome design were tested and it was found that charged and mannosylated nanoliposomes had enhanced uptake and antibacterial activity [76]. The following year, the antibiotic, rifampicin, was loaded into mannosylated SLNs and administered to mice as an intratracheal powder aerosol [66]. It was similarly found that mannosylation of the LNPs enhanced antibiotic uptake in alveolar macrophages obtained from mouse lung tissue sections, supporting the 2019 finding [66,76]. Furthermore, poor systemic distribution of the SLNs to mouse extra-pulmonary organs was observed, suggesting a possible reduction in systemic toxicity and improved lung retention of inhaled drugs [66]. Another study demonstrated the encapsulation of the slightly water-soluble wide-spectrum antibiotic ofloxacin in SLNs [78]. In dialysis membrane experiments where solvent ofloxacin concentration was determined at intervals, the SLN-formulated ofloxacin exhibited sustained release of encapsulated ofloxacin for 24 hours [78]. SLN–ofloxacin also decreased the minimum inhibitory concentration against airway-infecting *Pseudomonas aeruginosa* bacteria that often plagues CF patients by 6.1- to 16.1-fold [78]. When administered to mice as a dry powder through nose-only inhalation, ofloxacin reached therapeutic concentrations in the mouse lung interstitial fluid while staying below the toxic levels [78].

Treating CF requires improving the ion channel activity of the CF transmembrane conductance regulator (CFTR) protein and reducing lesions [79]. Drugs such as lumacaftor and ivacaftor accomplish this by correcting a deletion mutation in CFTR and increasing the frequency of CFTR taking on an open conformation, respectively [79]. Garbuzenko et al. in 2019 found that delivering these two hydrophobic drugs within a nebulized NLC formulation restored the activity of CFTR and reduced the volume of fibrotic tissue present in the treated mouse lungs, thereby treating CF effectively in the murine model [79].

Anticancer drugs have also been packaged within LNPs to enhance local delivery to the tumour site while limiting systemic exposure to cytotoxic chemotherapy drugs [64,73,74,80]. The in vitro efficacy of lipid-encapsulated anticancer drugs was demonstrated on cultured A549 human lung cancer cells [81]. Loading gefitinib, an epidermal growth factor receptor tyrosine kinase inhibitor, within SLNs resulted in better cellular uptake of the drug and subsequently a greater cytotoxic effect [81]. Recent rat and mouse studies further support the use of LNPs to deliver chemotherapeutics directly to the tumour site in the lungs [73,74,80]. Paclitaxel, triptolide and silibinin encapsulated within either aerosolized liposomes or SLNs were endotracheally administered to rat and mouse models of non-small cell lung cancer (NSCLC) [73,74,80]. They exhibited enhanced drug bioavailability in the lungs and greater anticancer efficacy while reducing systemic distribution to non-target organs compared with their naked drug counterparts or intravenously administered LNP formulations [73,74,80]. Pulmonary delivery of chemotherapeutics, as opposed to intravenous administration, was shown to reduce the dependence on tumour vascularization, which can often be poor in lung tumours, and limit drug concentrations at the target site [74]. Innovation in LNP surface chemistry also contributed to their superior efficacy [73,74]. Targeting moieties, such as antibodies against carbonic anhydrase IX, expressed aberrantly on lung cancer cells, and CPP33 peptide, with the ability to penetrate human NSCLC cell membranes, was investigated [73]. Modifying triptolide-loaded liposomes with both surface ligands further enhanced tissue penetration, drug uptake and cytotoxic activity in 3D spheroids of human A549 lung cancer cells and mice orthotopic tumour models [73]. Nevertheless, penetrating the pathological mucus can be challenging and impose limits on the benefits of inhaled nanomedicine [74]. Rosière et al. showed that this could potentially be overcome by conjugating chitosan residues to the liposomal surface [74]. Chitosan is a biocompatible carbohydrate polymer that gets protonated in acidic formulations, becoming positively charged and boosting LNP penetration through negatively charged mucins in the airway mucosal layer [74]. This could translate into prolonged residence in the lungs by overcoming mucociliary clearance mechanisms [14,21].

The treatment of other chronic lung diseases such as LAM and PAH could also benefit from the development of inhaled LNP technology [1,82,83,84,85,86]. LAM is a lung disease where the overly active mammalian target of rapamycin (mTOR) pathway causes the pathological uncontrolled proliferation of airway smooth-muscle-like cells [83]. This disease is currently treated with orally delivered free rapamycin in order to inhibit mTOR [83]. A 2020 study showed that encapsulating rapamycin in SLNs offered more efficient transport of the drug across Calu-3 bronchial epithelial cells in in vitro air–liquid interface experiments [83]. The nanoparticles additionally caused reduced proliferation of tuberous sclerosis complex-2 negative LAM-like cells at lower rapamycin doses than the free drug [83]. This could help increase bioavailability and minimize the side effects induced by higher doses of oral rapamycin. PAH, characterized by elevated blood pressure in the pulmonary arteries, is largely treated with vasodilators to reduce the blood pressure [84]. In 2018, three studies investigated the prospect of encapsulating vasodilators treprostinil and sildenafil within LNPs for local pulmonary delivery via nebulized inhalation delivery [82,84,86]. These studies observed that LNP encapsulation enabled sustained drug release in the lungs and reduced systemic exposure over the course of treatment in rat in vivo experiments [82,84,86]. The increase in right ventricular pulse pressure was also inhibited in dogs and rats, with a 60-fold reduction in plasma treprostinil concentration at 50% inhibition of vasoconstriction [82]. Interestingly, loading the prodrug form of treprostinil into the LNPs further aided the sustained vasodilation effects as additional time was needed for its dissociation from the LNP and conversion into active treprostinil by target cell enzymes [82,84]. Such a mechanism can prevent the pulmonary tissue from being exposed to a spike of vasodilators and thus reduce excessive vasodilation that can often lead to side effects such as alveolar bleeding [84,86]. Moreover, this has the potential to reduce the required dosing frequency if translated to human disease treatment, making regimen adherence more convenient [82,84].

Tuberculosis (TB) remains a major public health concern, it is caused by lung infection of *Mycobacterium tuberculosis* (Mtb). As Mtb resides inside the alveolar macrophages, native antibiotic drugs have difficulty crossing the plasma membrane barrier and reaching the macrophage intracellular region, thus reducing the efficacy of TB treatment. Hence, anti-TB drugs need to be efficiently delivered to the intracellular regions of macrophages. Inhaled LNP packaged anti-TB drugs offer an excellent avenue to improve drug availability within macrophages. Both passive targeting of alveolar macrophages utilizing receptor-independent uptake pathways and active targeting by incorporating a macrophage-specific cell surface receptor have been explored. This area has been recently reviewed in detail and we will not cover it here [87].

The abundance of evidence demonstrates improved drug performance, more favourable pharmacokinetic and biodistribution profiles as well as enhanced tolerance of side effects. Thus, LNPs as a pulmonary drug delivery system offer many advantages for the treatment of chronic and acute lung diseases. With the plethora of small molecule drugs used in clinics today, many can be adapted for LNP encapsulation, either in their active or prodrug forms.

### 3.2. Nucleic Acids Drugs

Nucleic acid therapeutics have also been on the rise in recent years with improvements in delivery formulations [88,89]. The ability to provide the patient’s target cells with the genetic information to produce a desired protein, or to knockdown the expression of a harmful protein can bypass the need to formulate a drug around the individual protein’s chemistry [88]. There are, however, a few hurdles that negatively charged nucleic acids such as mRNA, siRNA and antisense oligonucleotides must overcome to function as drugs effectively. These challenges include the need for these drugs to be protected from nuclease degradation on their journey to target cells, to interact with negatively charged cell membranes during endocytosis and subsequently escape the endosome to reach the cytosol or the nucleus [34,52,90]. LNPs provide an ideal encapsulation system to overcome these challenges [28,33,34,90]. The ionizable cationic lipid components of LNPs are conducive for membrane interaction and endosomal escape [27,34,35]. The first siRNA drug, ONPATTRO^®^ (Alnylam Pharmaceuticals Inc.), which was approved by FDA in 2018 for treating hereditary transthyretin-mediated amyloidosis, contains the ionizable lipid MC3 in its LNP formulation [34,35,36,91]. This helped to spur the development of more lipid candidates for newer LNP–nucleic acid therapeutics [34,52]. The intramuscular COVID-19 mRNA vaccines produced by Pfizer–BioNTech and Moderna employ liposomes formulated with the ionizable lipids ALC-0315 and SM-102, respectively [28,40,92,93,94]. The progressively acidic environment of the endosome facilitates their protonation and fusion with the negatively charged endosome membrane, releasing the nucleic acid payload into the cytoplasm [34,35]. The success of the liposomal mRNA COVID-19 vaccines in the current pandemic has advocated for the safety and efficacy of LNP delivery technology in nucleic acid therapeutics [35,95].

A 2020 study demonstrated that LNP-mediated pulmonary delivery of mRNA was capable of lung specificity while protecting the mRNA payload [90]. mRNA encoding the firefly luciferase reporter enzyme was encapsulated in LNPs and administered intratracheally to mice using the Aerogen^®^ Solo nebuliser (Galway, Ireland). Luciferase expression was detected in the mouse lungs six hours after instillation whereas negligible expression was elicited in the heart, liver and kidneys [90]. In comparison, naked luciferase mRNA failed to be delivered effectively to the lung tissue and no protein expression was achieved [90]. This result supported the use of inhaled LNPs to protect and deliver mRNA to the lungs to generate desired protein expression.

The use of inhaled LNPs to deliver nucleic acid drugs in treating respiratory diseases has been reported [96,97]. Mice were exposed to the nebulized vapour of LNP-encapsulated mRNA encoding for an anti-haemagglutinin neutralizing antibody via nose-only inhalation [97]. Exposed mice were protected against an H_1_N_1_ influenza A viral infection compared with the control mice [97]. This showcased LNPs’ ability to facilitate pulmonary mRNA delivery, uptake of mRNA and downstream protein expression to induce a therapeutic effect in vivo. In 2018, Robinson and co-workers investigated a similar system to treat chronic inflammatory lung diseases like CF [63]. Their approach focused on restoring the expression of the faulty CFTR protein in mice after spontaneously inhaling a LNP–mRNA formulation via nasal instillation [63]. When CFTR-knockout mice inhaled LNPs carrying CFTR mRNA, they observed CFTR activity and chloride secretion function restored up to 55% [63]. This could translate to an alleviation of the thick mucus and a reduction in lung bacterial infections in human CF patients. The mRNA therapeutics firm Translate Bio developed LNPs to carry CFTR mRNA (MRT5005) into the lungs of human patients [34]. Translate Bio’s MRT5005 is currently in a Phase 1/2 first-in-human trial (NCT03375047) to ascertain the safety of nebulized doses of MRT5005, making this the first inhaled mRNA therapeutic to be trialled [34,98]. More recently, an inhalation delivered aerosolized LNP-encapsulated mRNA encoding for the protein DNAI1 demonstrated effective delivery of DNAI1 mRNA into the lower airways and the effective enhancement of ciliary beat frequency in an inducible Dnaic1 knockout mouse model for primary ciliary dyskinesia (PCD) [99]. PCD is a respiratory disease caused by dysfunction of the cilia with currently no approved treatments. DNAI1 is one of the more frequently mutated genes involved in cilia function. This study suggested the potential for an mRNA therapeutic to correct cilia function in patients with PCD due to DNAI1 mutations.

Treating CF could also be mediated by inhaled LNP–siRNA as demonstrated by Tagalakis and colleagues [100]. The epithelial sodium channel (ENaC) often becomes overexpressed and hyperactive following the loss of functional CFTR protein [100]. Receptor-targeted cationic liposomes carrying siRNA against the essential alpha subunit of ENaC were administered to mice via oropharyngeal instillation. A single dose caused a 30% reduction in ENaC expression and this increased to 50% following a triple dose, where the silencing effect persisted for a week post dose [100]. Most recently, cationic liposomes were again shown to effectively deliver siRNA to mouse lungs via intratracheal injection to silence sushi repeat-containing protein X-linked 2 (Srpx2) [101]. Srpx2 is crucial to the progression of scarring in idiopathic pulmonary fibrosis, and a reduction in its expression can protect the lungs against further fibrosis [101]. LNPs thus offer a way to protect naked nucleic acids from degradation, improve their stability, hasten translocation across mucus and enhance local lung concentration of the therapeutic. All this while limiting systemic exposure and without eliciting immune responses, making this a promising modality to therapeutically alter protein expression levels for chronic respiratory diseases [34,90,96,97,100].

### 3.3. Protein and Peptide Drugs

In addition to small molecules and nucleic acid drugs, various protein/peptide-based drugs are also being developed. These include antibodies, enzymes, hormone mimics, growth factors, vaccines, neurotransmitters and ion channel ligands for analgesic, cardiovascular, oncology, metabolic and respiratory applications [102]. Protein/peptide drugs can inhibit protein–protein interactions and bind to their targets with more specificity than small molecule drugs [102,103]. Though chemically similar, amino acid chains smaller than 5 kDa or shorter than 40–50 amino acids are typically classified as peptides whereas larger chains are considered proteins [102,103]. This size difference confers upon them different properties. For example, smaller peptide drugs are cheaper to produce, exhibit greater tissue penetration and are less immunogenic but have a smaller surface for target interaction than larger protein drugs [102,103]. Protein and peptide therapeutics often face significant risks of degradation and low intestinal absorption when administered orally [41,104]. Hence, most are administered through intravenous infusions or intramuscular injections but repeated treatments are required and the procedure is invasive and inconvenient [41]. Some inhaled non-encapsulated protein and peptide drugs include monoclonal antibodies, granulocyte-macrophage colony-stimulating factor protein for pulmonary alveolar proteinosis, neutrophil elastase inhibitor Alpha-1-antitrypsin, deoxyribonuclease I (Pulmozyme^®^, Genentech Inc., South San Francisco, CA, USA) and ENaC inhibitor SPLUNC1-derived peptide for CF [103,105,106,107]. However, the presence of proteases in the airways can threaten the structural integrity of inhaled protein therapeutics [14,108].

Pulmonary delivery combined with LNP encapsulation can mitigate these downsides [102,103]. LNP encapsulation is the most advanced formulation in clinical development for pulmonary delivery and can further improve the performance of inhaled peptide-based drugs. However, there has been a smaller volume of research pertaining to inhaled protein and peptide drugs packaged within LNP delivery systems compared with small molecule drugs and nucleic acid payloads [104,105,109]. In the existing literature, more investigations have been conducted on inhaled LNP-encapsulated peptide drugs intended for subsequent systemic circulation following pulmonary delivery [110]. Examples include systemic delivery of peptide hormones insulin and calcitonin, as well as the derivatized peptide exendin-4 [110,111,112]. Insulin in particular has been the subject of inhalation therapy research for some time [41]. When SLNs carrying insulin were administered to rats via nose-only inhalation of an aerosol cloud or through intratracheal instillation, they produced good distribution throughout the lung alveoli and a sustained hypoglycaemic effect [41,113,114]. Although the two approved inhaled insulin drugs for systemic delivery, Exubera^®^ (Pfizer Inc., New York, NY, USA) and Afrezza^®^ (Mannkind Corporation, Westlake Village, CA, USA), did not perform well commercially, there is potential for other inhaled LNP–proteins/peptides targeting lung diseases [115].

Inhaled liposomal vasoactive intestinal peptide (VIP) is an example of an LNP-packaged peptide drug targeted towards pulmonary disease treatment. VIP is a 28-amino acid-long peptide first identified from the small intestine. Its receptors are present in the airway smooth muscle tissue as well as in basal portions of the lung mucosa [116,117]. VIP has been observed to possess bronchodilator and vasodilator effects, pointing to its possible utility in asthma and PAH treatment [116,117]. A study probed the effects of encapsulating and nebulizing VIP within liposome carriers [116]. Following nebulization from the Micro Drop Master Jet^®^ (MPV Truma, Putzbrunn, Germany), VIP-loaded liposomes displayed significantly enhanced artery relaxation compared with free VIP in ex vivo experiments on excised rat pulmonary arteries [116]. Despite the low number of research examples on inhaled protein therapeutics in the current literature, we anticipate that the research interest in inhaled proteins and peptides may soon be accelerated by the showcased efficacy in LNP encapsulation technology. A summary of inhaled therapeutics for respiratory diseases that have been investigated in animal models is presented in Figure 2.

## 4. Advantages of Inhaled LNP Therapeutics

### 4.1. Diverse Drug Compounds Delivered

The diversity in possible LNP chemistry and structure allows a versatile carrier system. For instance, liposomes are composed of an amphiphatic lipid membrane and an aqueous core, whereas SLNs and NLCs possess a hydrophobic lipid core containing glycerides, esters and waxes [30,39]. Both hydrophilic and hydrophobic therapeutics can be loaded, possibly in combination, with the available LNPs [62]. This is especially important for many newly developed small molecule drugs as their targets are increasingly embedded within hydrophobic environments such as cell membranes. This makes the drugs more likely to be hydrophobic themselves [21,71,72,79]. Chemotherapeutics are especially likely to be hydrophobic, which limits systemic bioavailability when administered as free drugs [21,64,72]. The nanometre sizing of LNPs also confers upon them a high surface area to volume ratio [21,23]. This additionally promotes the dissolving of LNPs into the aqueous extracellular environment of the respiratory tract [21,23]. Charged payloads such as nucleic acids can also be delivered by liposomes, where the ionisable cationic lipid component promotes membrane interaction and endosomal escape [27,34,35]. LNPs could therefore greatly contribute to many areas of lung disease treatment as they provide a modality to improve the delivery of both established and new drugs to the lungs.

### 4.2. Protection from Degradation and Improved Drug Stability

Administering naked, unprotected drugs makes them susceptible to degradation and packaging them within drug carrier systems such as LNPs for inhalation therapy would mitigate these losses [21,34,39,90]. By exploiting the pulmonary route, drug loss to gastrointestinal degradation and first-pass metabolism in the liver can be avoided [20,62,118]. However, the pulmonary tract is also host to many xenobiotic metabolizing enzymes that can alter the integrity and pharmacokinetics of inhaled therapeutics [108,119]. These include cytochrome P450 (CYP) oxidoreductases, hydrolases, glutathione S-transferases, UDP-glucuronosyltransferases, sulfotransferases, N-acetyltransferases and proteases [14,108]. Members of the CYP superfamily of enzymes are expressed in the bronchiolar epithelium as well as in the alveolar cells and alveolar macrophages [108]. The nasal epithelium also possesses metabolic activity due to NADPH-CYP expression [108]. They are key to the metabolism and fate of inhaled small molecule drugs, having implications for drug efficacy. For example, CYP3A4 and CYP3A5 are the predominant forms of CYP3A expressed in human lungs, responsible for metabolising the inhaled β2-adrenergic agonist salmeterol and ICS fluticasone propionate [108,119,120,121,122]. Pulmonary protease and peptidase activity pose a challenge to delivering naked protein/peptide drugs via inhalation, with protease inhibitors demonstrating a positive effect on drug bioavailability [14,108]. Additionally, administering drugs via the pulmonary route also demands that the drugs survive the shear forces exerted upon them during the aerosolization process [103,105]. Protecting inhaled therapeutics within nanoparticle shells such as LNPs serves to delay degradation, increase the bioavailability of intact compounds, and improve pharmacokinetic profiles [14,62,108,123,124].

### 4.3. Enhanced Drug Retention and Reduced Systemic Toxicity

On top of surviving in vivo degradation, therapeutic delivery via the airways using LNPs as the delivery platform continues to showcase benefits. Larger fractions of the administered dose can reach the lungs and expose the diseased tissue to higher concentrations of the needed therapeutic [20,125]. It was demonstrated that an inhaled dose of 100–200 µg of salbutamol was equivalent to a 20-fold higher oral dose of 2–4 mg [20]. Moreover, LNP encapsulation increased pulmonary drug retention times [21,62,124]. The processes of airway cellular uptake via endocytosis or dissociation of the therapeutic compound from LNP carriers further contribute to the delayed and controlled release of the active therapeutic [21,62,67,82,84,86,124]. Drug activity can therefore be sustained for a longer duration between doses. A similar therapeutic effect can be achieved with a lower administered dose, sparing the patient from unnecessarily high doses and the subsequent systemic toxicity [20,82,84,90,100].

### 4.4. Reduced Immunogenicity

The use of bio-similar phospholipids, cationic or ionizable lipids and cholesterol in LNP formulations reduces the likelihood of inducing an immune response [27,62,96]. They are physiologically compatible as the main phospholipid components used in LNPs (e.g., DPPC and DSPC) are naturally occurring in human lung surfactant fluid [126,127]. LNPs shield the payload from the immune system recognizing it as foreign and reduce phagocytic clearance by alveolar macrophages [62,103]. Patients can gain a more rapid clinical response and lower dosing frequency with less severe side effects. This would help maintain patient compliance and achieve better outcomes [20,21,124,128]. 

In the case of the PEGylated lipid component of LNP formulations, there have been reports demonstrating it can trigger an immune response [28]. Anti-PEG IgG and IgM antibodies have been observed in the blood samples of both individuals previously treated with PEGylated therapeutics and individuals with no prior exposure to such therapeutics [129,130,131]. This has been associated with the occurrence of accelerated blood clearance and hypersensitivity reactions of circulating PEGylated drug carriers such as liposomes and SLNs [130,132]. Potential antigens circulating in the blood are also more likely to be captured by secondary lymphoid organs and be presented to the adaptive immune system for further memory formation against them, rendering repeat treatments of PEGylated LNPs less effective [133]. In contrast, the respiratory tract is a mucosal surface that harbours reduced IgG and IgM antibodies, as well as being protected largely by the innate immune system instead [134,135,136,137,138,139]. Inhaling LNP drugs offers a strategy to reduce their propensity to trigger an immune reaction compared with administering unprotected drugs or administering them systemically.

## 5. Challenges of Inhaled LNP Therapeutics and Design Considerations

### 5.1. Overcoming Physiological Barriers in the Lungs

#### 5.1.1. Airway Structure

Although the large surface area in the airways offers a promising target for drug delivery and absorption, the structure of the respiratory tract is heterogeneous and can impact drug delivery [140]. The airways are divided into two main zones: the upper conducting zone and the lower respiratory zone [39,140]. The upper conducting airways encompass the nose, oral cavity, pharynx, larynx, trachea and its first few bifurcations into bronchi and bronchioles [14,39]. The lower respiratory zone comprises the terminal bronchioles leading to the alveolar ducts as well as the alveoli, sites of gaseous exchange residing deep in the lung tissue [14,39]. There are many opportunities for inhaled droplets to be impacted in the airways before reaching the deeper lung tissue [14]. Resident cells lining the different zones differ and contribute to differences in drug uptake. The ciliated, mucus-secreting epithelial cells of the upper conducting zones pose a greater barrier to drug uptake than the thinner epithelium cell layer lining the bronchioles and alveoli in the lower respiratory zone [39]. Hence, the impact of airway structure on particle deposition is an important factor in inhaled LNP design for the effective delivery of therapeutics to the target site. 

#### 5.1.2. Airway Clearance Mechanisms

Inhaled particles also must overcome clearance mechanisms utilized by the lungs to protect its mucosal layer from pollutants, dirt and pathogens. These can be categorized into physical and immunological clearance mechanisms or barriers [14,39,140].

The physical barrier is located on the airway surface layer in the bronchial region [14,21]. It is composed of a luminal mucus gel layer and a periciliary liquid layer where ciliated cells beat hair-like cilia structures [14,21]. The coordinated cilia movement forms the mucociliary escalator that moves mucus-trapped particles from the conducting airways upwards to the nasopharynx and oropharynx [14,140,141]. Subsequently, potential pathogens or beneficial therapeutics may be expelled via coughing or ingestion into the gastrointestinal tract for potential destruction [140,141]. The properties of pathological mucus in diseased lungs also often pose an increased challenge. Pathological mucus is more dehydrated, more viscous, and thus contains a higher concentration of glycosylated mucin proteins [21,140]. Glycosylated mucin proteins contain numerous sialic acid and sulphur residues, giving the network a strong negative charge [141]. The resulting mesh of proteins clusters tighter together and forms narrower pores in pathological mucus, making it harder for LNP therapeutics to penetrate [21].

Immunological clearance mechanisms refer to the lung-resident cells of the innate immune system that survey the tissue for foreign particles [14,62,134,135]. The main innate immune cells of the lung are the alveolar macrophages residing in the airways of the alveoli [62,135]. Nanoparticle therapeutics would encounter the alveolar macrophages after bypassing the physical barriers of the upper conducting airways [62]. Alveolar macrophages are professional phagocytes that internalize foreign particulates and digest them within lysozyme-filled vesicles, transporting them to the lower entrance of the mucociliary escalator or to the lymph system for clearance [14,39,62]. In most cases of inhaled LNP delivery to treat lung diseases, the aim would be to avoid alveolar macrophage uptake as much as possible. However, the macrophages themselves can be the target cells in certain applications, exemplified by the treatment of *Mycobacterium tuberculosis* bacteria residing within the macrophages [62,66,76]. Therefore, the nature of the disease is also important in determining the desired interaction with pulmonary physiological barriers during drug development.

### 5.2. Design Considerations for LNP Aerosols for Airway Delivery

#### 5.2.1. Formulating the Appropriate Particle Size

Different respiratory diseases impact different positions of the airway. For example, asthma mainly affects the upper airways whereas COPD mainly affects the distal alveoli. To successfully deliver a drug to the desired location, particle size is a most important parameter that can determine the region of the airways in which LNP deposition occurs and the success rate of bypassing the physiological barriers and clearance mechanisms.

Inhaled matter deposition in the airway is influenced by multiple mechanisms with the aerodynamic diameter (AD) of an inhaled droplet or particle largely determining the deposition pattern [18,62,140,142]. Larger particles with an AD larger than 5 μm are likely to undergo inertial impaction in the upper and large airways [18,62,140]. This often causes particles to settle in the oropharyngeal region, being at risk of entering the gastrointestinal tract instead of the airways [18,103]. If the particle AD is between 1 and 5 μm, gravitation sedimentation occurs leading to deposition in the lower airways, in the terminating bronchioles and alveoli for deep lung tissue penetration [18,62,103,140]. Finally, particles in the nanoscale with AD below 1 μm can access the alveoli deep in the lungs and are subjected to random diffusion via Brownian motion [39,140] (Figure 3). Such diffusion confers both advantages and drawbacks. On one hand, these ultrafine particles can diffuse across not only the lower alveolar region but also the conducting tracheobronchial region [39]. This promotes widespread distribution of LNP-encapsulated payloads throughout the lungs and would be particularly beneficial for spatially distant target cells such as lung tumour growths dispersed throughout the lung tissue [74]. On the other hand, particles this small are liable to be exhaled out of the airway prior to deposition, especially if the patient does not sufficiently practice breath-holding upon dosing from the inhaler device [18,62,103,140,143].

A distribution of micro-scale LNP aggregates with an AD between 0.5 and 5 μm has been described to distribute throughout the bronchial–alveolar region via gravitation sedimentation and Brownian motion [18,39,62,140]. Subsequently, the moisture of the airway surface layer would encourage the dissolving of the carrier matrix, releasing smaller individual LNPs [18,143]. An AD range of 100 to 300 nm is typical for individual LNPs [23,41]. The details of each respiratory disease and therapeutic compound in each application need to be explored to determine the optimal size of the LNPs. For example, treating tuberculosis requires antibiotic delivery into the bacteria-infected alveolar macrophages residing deep in the alveoli. Slightly larger particles (200 to 300 nm) would encourage alveolar macrophage uptake and induce the desired anti-bacterial therapeutic effect [66,76]. In contrast, diseases with non-macrophage target may require individual LNPs with ADs smaller than 200 nm to avoid macrophage uptake and immunological clearance away from their desired target [62]. Careful study and design of individual LNP particles as well as aggregates is important to a successful formulation.

#### 5.2.2. Increasing Mucus Adhesion and Penetration

It is also paramount to develop strategies to overcome the luminal mucus gel layer and the mucociliary escalator clearance mechanisms [14,21,140,141]. Airway mucus is largely comprised of mucin proteins containing a high concentration of conjugated carbohydrate chains [141]. The presence of numerous sialic acid and sulphur residues within these chains makes the mucin proteins strongly negatively charged, repelling similarly charged particles [141]. Increasing adherence and penetration through the airway mucus would reduce clearance and potentially increase lung retention times. In addition to the cationic lipid component of LNPs, multiple surface chemistry modifications have been investigated [27,34,35].

Chitosan is one such polymer that confers mucoadhesive properties [62,74]. It is a polysaccharide composed of D-glucosamine and N-acetylglucosamine residues linked via β-1, 4-glycosidic bonds with multiple amine groups attached [74]. This chemical structure makes it biocompatible, non-toxic and biodegradable [62,74]. The multiple amine groups contribute to the positive charge of chitosan in the approximately neutral pH of airway mucus [74,141]. Electrostatic interactions with the negatively charged mucins give chitosan-coated particles the ability to adhere to the airway mucosal surfaces [74,144]. Chitosan-coated particles are also less likely to be carried in the mucociliary escalator [74,144]. Additionally, chitosan can negotiate itself between the tight junctions of airway surface epithelial cells, promoting drug uptake and bioavailability in the lungs [144,145]. There are also a variety of chitosan derivatives that can be utilized, depending on the specific chemistry of each LNP formulation and its demands, such as trimethyl chitosan, carboxymethyl chitosan and PEGylated chitosan [74]. Such variety can improve the versatility and breadth of the therapeutics that can be delivered by inhaled LNPs. 

PEGylation of LNPs is another surface modification that enhances airway mucus penetration (Figure 1D). The polymer of ethylene glycol is hydrophilic yet neutrally charged and was utilized to mimic the capsid coatings of mucus-penetrating viruses without the risk of pathogen-mediated immunogenicity [39,141,146]. De Leo and colleagues performed in vitro experiments demonstrating the greater penetration ability of PEGylated liposomes in pathological sputum obtained from COPD patients [147]. This quality aids nanoparticles in evading mucociliary as well as immunological clearance by alveolar macrophages in the lungs [28,39,141,146]. Furthermore, PEGylation was shown to increase the solubility of hydrophobic therapeutics and stabilize coated liposomes [39,97,146]. In 2016, Luo et al. demonstrated that conjugating PEG to the anticancer drug paclitaxel led to a 3.33-fold increase in the percentage of retained drugs in mouse lungs 48 h after intratracheal administration over the free drug [146]. Therefore, when PEG is conjugated to the surface of nanomedicines, the drug’s pharmacodynamics is often improved [39,141,146,147]. Despite this, a measured application of PEGylation conjugation is required. Some teams observed that if the PEG coat density is too high, cellular uptake of the coated therapeutics can be compromised due to reduced contact with target cell membranes [39,141]. More investigations are needed to gauge the ideal coating parameters for each LNP formulation. However, the approval of PEG as an excipient for inhaled treatments suggests that PEGylation is a beneficial addition to the design of LNPs [39]. 

Additional coating molecules that penetrate airway mucus include poloxamers, peptides, poly(2-oxazoline) and N-(2-hydroxypropyl) methacrylamide [141,148,149]. Poloxamers are amphiphatic polymers of polyoxypropylene and polyoxyethylene that were shown to increase the diffusion rate of coated SLNs across artificial sputum media in vitro compared with uncoated SLNs [149]. Another study confirmed that poloxamer showed no cytotoxicity in cultured human alveolar epithelial cells, making it suitable for pulmonary applications [150]. A team in 2020 demonstrated that short bacteriophage heptapeptides screened from a phage display library also had the ability to improve particle transport across ex vivo CF mucus [148]. Taken together, altering the surface of LNPs can help them overcome clearance mechanisms, reside longer within the airways, and release their payloads over a more sustained duration, therefore increasing local drug concentration and the therapeutic effect [39,62,65,67,74,141,146]. These advancements help expand the design options in formulating more optimal inhaled LNPs for treating respiratory diseases in the future.

### 5.3. Selection of Inhaler Devices and Improving Their Compatibility

In addition to the efficacy of the drug formulations themselves, potential LNP drugs must also be compatible with the currently available inhaler devices in order to meaningfully translate to clinical use [151]. There is a range of devices currently available such as dry powder inhalers (DPIs) for dry powder formulations, and pressurized metered-dose inhalers (pMDIs), soft mist inhalers (SMIs) and nebulizers for liquid formulations [14,18,39,103,151]. They all possess individual advantages and disadvantages. pMDIs and DPIs have been widely used to deliver SABAs, LABAs and LAMAs for the treatment of obstructive lung diseases such as COPD and asthma [18,151].

Being compact and portable, pMDIs are also able to deliver a wide range of drugs in solution using propellants, even to intubated patients [18]. The drawbacks of pMDIs include the need for the patient to possess adequate hand–breath coordination, where inconsistency in technique can affect dosing and therapeutic outcomes [18,151]. Furthermore, the liquid conditions in pMDIs and nebulizers increase the tendency of premature protein drug degradation while the shear forces exerted during nebulization can disrupt certain liposome bilayer structures [62,103,105,152]. However, Zhang et al. described the retention of firefly luciferase reporter expression before and after nebulization of LNPs carrying the corresponding mRNA [90]. Additionally, the increasingly popular next-generation LNPs (SLNs and NLCs) may possess improved stability and tolerance against shear forces [86]. These allow liquid formulations and their accompanying inhaler devices to remain relevant in the future development of inhaled LNP drugs.

As the decision on which inhaler device is suitable for which patient, disease or medication is a multifactorial one, various studies have been conducted with the aim of increasing flexibility available to patients and healthcare providers. A study investigated a novel co-suspension delivery platform to enable the delivery of multiple LNP drugs simultaneously via pMDIs [18,151,153]. Micronized drug crystals and spray-dried porous phospholipid particles were suspended in hydrofluoroalkane propellant [153]. Both in vitro and in vivo experiments displayed this technology’s ability to maintain drug stability while providing consistent drug delivery and lung deposition in healthy human volunteers [18,151]. Moreover, this held true under changes in flow rate and usage patterns [18,151]. Patients suffering from multiple respiratory diseases would benefit from reliable dosing of multiple medications from a single pMDI device [18,153]. Since the pMDI is the most popular inhaler device, further improvements on suspension technology would therefore have a great impact on respiratory disease treatment and management. 

In contrast, patient hand–breath coordination is not required to dispense the intended dose from a DPI. Propellants are also not required in the DPI-loaded drugs. Hence, there is no need to consider the drug’s compatibility with the propellants in the formulation [151]. Liposomal drug stability has also improved with the use of dry powder formulations in DPIs, contributing to their increasing popularity [14,39,103,154]. However, patients have to generate a sufficiently forceful intake of breath to effectively receive the prescribed drug dose [14]. This can prove challenging for certain patient populations including young children, the elderly and those with severe progression of obstructive lung diseases [14,19,151]. DPIs may thus be more suitable for administering maintenance therapy to patients with a manageable disease stage and adequate lung function. Patients with more limited inhalation ability may need to opt for alternative devices or first be treated with other rescue therapeutics [14].

Dry powder LNP drugs have also been advancing in recent years to develop more therapeutics compatible with DPIs [110,155]. Liposomal drugs in dry powder form not only reproduce drug activity but show improved drug stability during storage over their nebulized liquid and solid equivalents [62,103,105,154,156,157]. Secretory leukocyte protease inhibitor indicated for inflammatory lung diseases was encapsulated in liposomes, lyophilized and then micronized to produce an inhalable dry powder containing microparticles containing LNPs suitable for pulmonary delivery [154]. Doing so also reduced drug inactivation by cathepsin L in vitro over an aqueous formula [154]. The antioxidant n-acetylcysteine (NAC), when embedded within liposomes in dry powder form, displayed greater in vitro antioxidant activity in rat lung homogenates over non-dry powder liposomal NAC and free NAC formulations [157]. The antibiotic ciprofloxacin used in non-CF bronchiectasis and COPD-derived infections caused by *Pseudomonas aeruginosa* was similarly formulated into dry powder liposomes [156]. It demonstrated its suitability for inhalation delivery and sustained drug release in in vitro aerosol and release assays [156]. With liposomes forming the majority of currently researched LNP variants, progress in dry powder liposomal formulations is expected to fuel drug development.

Diversification in excipient compounds can also greatly aid in the future development of stable inhaled LNP therapeutics. The scarcity of FDA-approved excipients for inhalation therapeutics is currently limiting drug development [103]. Liquid formulations often employ buffers and surfactants as excipients (for nebulizers, pMDIs and SMIs) whereas dry powder formulations frequently utilize sugars, polyols and amino acids (for DPIs) [103]. Increased toxicological testing of excipients would be beneficial. Additionally, toxicology tests aimed at the pulmonary administration route need to cover particle morphology, size distribution, surface chemistry and agglomeration tendency, which goes beyond classical toxicology tests focused on drug dose and composition [26]. Steiner et al. recently incorporated three orally-approved excipients: Kolliphor RH40Poloxamer 188 and Tween 80, into drug-loaded SLNs in vitro [150]. These particles demonstrated low toxicity and suitability for pulmonary applications in cultured human alveolar epithelial cells and macrophages [150]. Recent innovations in lactose monohydrate, the popular sugar-based carrier material in dry powder medications, were made as well. Flower-shaped lactose was developed to be loaded alongside curcumin-SLN powder [158]. It demonstrated a higher level of safety among in vitro cultured human lung epithelial cells than original lactose monohydrate, potentially making it the next efficacious excipient carrier for pulmonary delivery [158].

Production methods have also advanced with a recent study on a promising thin-film freeze-drying process for preparing inhaled dry powder SLN siRNAs [155]. Although spray-drying, lyophilization and new methods of freeze-drying are presently exciting opportunities for drug development, they can still be challenging to execute and scale up [156]. With the wide range of drug classes compatible with dry powder LNP formulations, further improvements to the preparation technology would greatly benefit the treatment of lung diseases. A summary of the advantages and challenges of LNPs for inhaled drug delivery is shown in Table 1.

## 6. Inhalation LNP Drugs Approved and in Clinical Trials

Table 2 summarizes the inhalation LNP drugs currently approved by the FDA or in clinical trials. Curosurf^®^ (Chiesi Farmaceutici S.p.A., Parma, Italy) is a liposome-encapsulated surfactant protein B and C (SP-B and SP-C) used for the treatment of respiratory distress syndrome in premature infants that was approved by the FDA in 1999. However, it is delivered by endotracheopulmonary instillation, not really an inhaled drug in strict terms [159,160,161]. In 2013, TOBI^®^ Podhaler^®^ (Novartis, Basel, Switzerland), a dry powder inhalation form of the antibiotic, tobramycin, developed by Novartis using its PulmoSphere^®^ technology (Novartis, Basel, Switzerland), was approved by FDA for persons with CF and chronic *Pseudomonas aeruginosa* lung infections [162,163]. In TOBI^®^ Podhaler^®^, the drug tobramycin is delivered as spherical porous particles around 1–5 µm in size with a mainly phospholipid surface via a DPI. Hence, strictly speaking, TOBI^®^ Podhaler is not a true LNP drug due to its relatively large particle size. Nevertheless, the development of TOBI^®^ Podhaler marked a significant milestone in the advancement of inhalation drug delivery.

In 2018, the FDA approved the first true inhalation LNP drug ARIKAYCE (amikacin liposome inhalation suspension). ARIKAYCE is an inhaled LNP encapsulated antibiotic for the treatment of non-tuberculous mycobacterial lung infections caused by Mycobacterium avium complex (MAC) in adults. ARIKAYCE is delivered via a nebulizer as part of a combination antibacterial drug regimen for adult patients who have limited or no alternative treatment options. As it is approved under the accelerated approval program with limited clinical safety and effectiveness data, ARIKAYCE is currently approved only for patients who do not achieve negative sputum cultures after a minimum of 6 consecutive months of a multidrug background regimen therapy [164]. In 2021, a frontline clinical trial program for ARIKAYCE was initiated to fulfil the FDA’s post-marketing requirement in order to gain full approval in the U.S. and to support a supplemental new drug application (sNDA) for its use as a frontline treatment for patients with MAC lung disease [70].

For drugs that have gone through or are currently in clinical trials, two concurrent phase III international clinical trials have been conducted with inhaled Pulmaquin (later named Apulmiq or Linhaliq in Europe). Apulmiq is a dual-release formulation composed of a mixture of liposome encapsulated and unencapsulated antibiotic ciprofloxacin (ARD-3150). Two phase III clinical trials were conducted for chronic lung infections with *Pseudomonas aeruginosa* in patients with non-CF bronchiectasis [165,166] (ORBIT-3 NCT01515007 and ORBIT-4 NCT02104245, Aradigm Corporation). In both studies, Apulmiq was safe and well tolerated. Although the ORBIT-4 trial led to a significantly longer median time to first pulmonary exacerbation compared with a placebo, ORBIT-3 or the pooled analysis of both trials did not give statistically significant results [165]. However, when combining data from both studies and only examining pulmonary exacerbations that were moderate or severe (i.e., those that required interventions with antibiotics or hospitalization), the median time to first pulmonary exacerbation in the Apulmiq group was significantly reduced. A subsequent post hoc analysis of the two identical trials revealed that inhaled ARD-3150 resulted in significant improvements in respiratory symptoms during the on-treatment periods that were lost during off-treatment periods [166]. Nevertheless, Apulmiq was rejected by the FDA, which requested the company to conduct a further better-controlled phase III clinical trial. In December 2020, Savara Pharmaceuticals discontinued the work on Apulmiq.

Dry aerosol inhalation delivery formulation of ciprofloxacin using the PulmoSphere^®^ technology (Ciprofloxacin DPI/BAYQ3939, Bayer/Novartis) has also gone through two phase III clinical trials for non-CF bronchiectasis [167,168,169]. Although the 14-day treatment cycle of the RESPIRE 1 trial showed significant improvements in prolonging the time to first exacerbation and reduced frequency of exacerbations compared with placebo, the 28-day treatment cycle in the same trial did not generate statistically significant improvements in the same primary endpoints. Meanwhile, a similarly designed RESPIRE 2 trial involving patients in different countries did not show statistically significant results. This dry powder aerosol delivery significantly improved the efficiency of drug deposition and ease of use compared with an earlier version using nebulizer delivered liquid formulation of the same drug. However, like TOBI^®^ Podhaler^®^, Ciprofloxacin DPI/BAYQ3939 also uses lipid microparticles, not true LNPs.

LNP-encapsulated CFTR mRNA (MRT5005) has also been developed and is currently in Phase I/II clinical trials by Translate Bio [34,98].

Altogether, inhaled LNP drug development is still in its early stages. Nevertheless, the many preclinical studies described in Section 2 and Section 3, with very promising results, clearly indicate the strong interest and great potential of this type of drug.

## 7. Conclusions

LNPs have come a long way from the first liposomes serving as fatty bases for topical creams and ointments. They have since been applied in systemic, then inhalation drug delivery, complemented by the development of newer particle types such as SLNs and NLCs. The pre-clinical studies in recent years have demonstrated the ability of LNPs to deliver various drugs, from chemotherapeutics, vasodilators, antibiotics, mRNA, siRNA and mucolytics, locally to the lungs for the treatment of pulmonary diseases such as lung cancer, obstructive lung diseases and microbial infections. Furthermore, LNPs achieved this while improving drug stability, reducing systemic toxicity and enabling higher local drug concentration in the lungs. This has been supported by the approval of inhaled LNP therapeutics such as Arikayce^®^ for treating Mycobacterium avium complex infections common in COPD, asthma, CF and bronchitis patients. Nevertheless, more thorough investigations are required to optimize LNP composition, particle size, formulation production methods, inhaler device compatibility and the safety profiles of more excipients. These efforts would help overcome the challenges of pulmonary delivery such as physiological barriers and the complex drug–patient interactions of different diseases. In conclusion, inhaled LNP therapeutics have great potential to play important roles in improving disease management and easing the global burden of various pulmonary diseases.

## Figures and Tables

**Figure 1 biomedicines-10-02179-f001:**
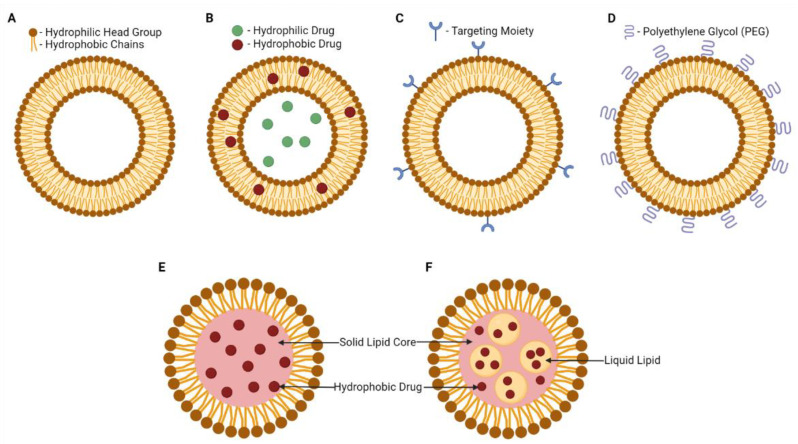
Visual illustration of the structures and composition of various LNPs. (**A**) Liposome. (**B**) Drug-loaded liposome. (**C**) Targeted liposome. (**D**) PEGylated liposome. (**E**) Solid lipid nanoparticle. (**F**) Nanostructured lipid carrier. Created with BioRender.com.

**Figure 2 biomedicines-10-02179-f002:**
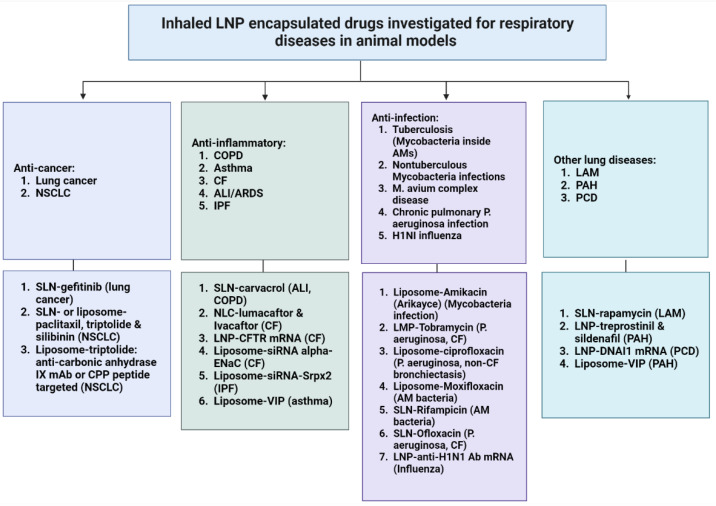
A summary of inhaled LNP-encapsulated drugs that have been investigated for respiratory diseases in animal models. Created with BioRender.com. ALI: Acute lung injury; ARDS: acute respiratory distress syndrome; COPD: chronic obstructive pulmonary disease; CF: Cystic fibrosis; IPF: idiopathic pulmonary fibrosis; LAM: lymphangioleiomyomatosis; LNP: lipid nanoparticle; NSCLC: non-small cell lung carcinoma; PAH: pulmonary arterial hypertension; PCD: primary ciliary dyskinesia; SLN: solid lipid nanoparticles; NLC: nanostructured lipid carriers; VIP: vasoactive intestinal peptide.

**Figure 3 biomedicines-10-02179-f003:**
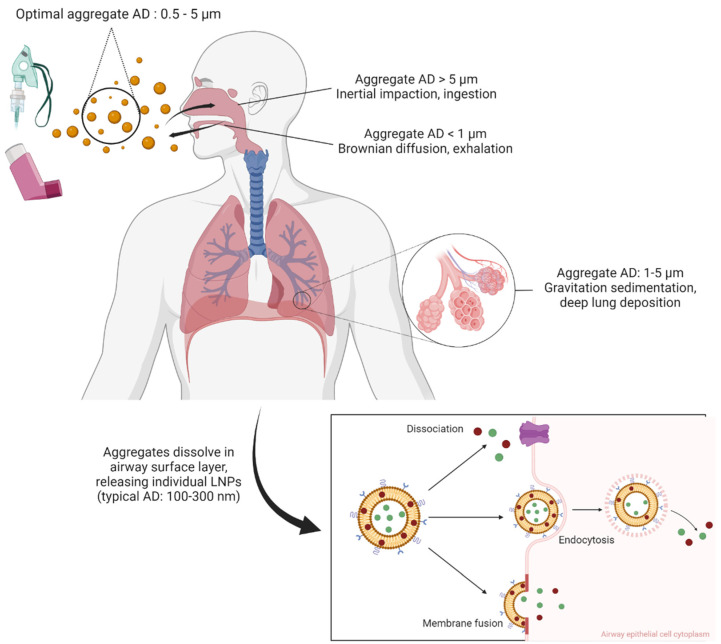
Schematic summary of AD-dependent deposition and distribution mechanisms of inhaled LNP aggregates in the airways. The mechanisms are inertial impaction, gravitation sedimentation and Brownian diffusion. Subsequent cellular uptake of individual LNPs and drug payloads by dissociation, endocytosis and membrane fusion facilitate therapeutic action (PEGylated and targeted liposome depicted). AD: aerodynamic diameter. Created with BioRender.com.

**Table 1 biomedicines-10-02179-t001:** Advantages and challenges of LNPs for inhaled therapeutics.

	Advantages	Challenges
1	Diverse drugs delivered	Airway clearance mechanisms
2	Improved drug stability	Airway structural barrier
3	Enhanced local drug retention	Premature particle deposition
4	Reduced systemic toxicity	Limited formulation excipients
5	Reduced immunogenicity	Inhaler selection and compatibility

**Table 2 biomedicines-10-02179-t002:** Inhaled or pulmonary delivered LNP drugs approved and in clinical trials.

Drug	Commercial Name	LNP Type	Disease	Status
SP-B and SP-C	Curosurf^®^	Liposome	Respiratory distress syndrome in premature infants	Approved 1999 [159,160].
Tobramycin	TOBI^®^ Podhaler^®^	LNP	Chronic pulmonary Pseudomonas aeruginosa (Pa) infection in CF patients.	Approved 2013 [162,163].
Amikacin	Arikayce^®^	Liposome	Mycobacterium avium complex lung disease	Approved 2018 [164].
Ciprofloxacin	Apulmiq (Linhaliq/ Pulmaquin)	Liposome	Chronic lung infections with pseudomonas aeruginosa with non-cystic fibrosis bronchiectasis	Two phase III clinical trials completed in 2016. Discontinued [165,166].
Ciprofloxacin	Ciprofloxacin DPI/ BAYQ3939	Lipid micro-particle	Non-cystic fibrosis bronchiectasis	Two phase III clinical trials completed in 2016. Not yet approved [167,168,169].
CFTR mRNA	MRT5005	LNP	CF	Phase I/II clinical trial [34,98].

Note: Arikayce is the only true inhalation LNP drug approved by FDA so far. CF: cystic fibrosis; CFTR: cystic fibrosis transmembrane receptor; LNP: lipid nanoparticle; SP-B and SP-C: surfactant-B and -C.

## Data Availability

Not applicable.

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
