# Peer review of "Lipid Nanoparticles as Delivery Vehicles for Inhaled Therapeutics"

_biomedicines, 2022, doi:10.3390/biomedicines10092179_

Round 1
Reviewer 1 Report
The review entitled “Lipid Nanoparticles as Delivery Vehicles for Inhaled Therapeutics” is well written and nicely summarize the use of lipid nanoparticles (LNPs) for inhaled therapeutics highlighting the successful attempts and limitations, referring to an extensive past and recent literature. It focuses the application on few most deadly lung diseases, describing the use of different formulations and structures of LNPs and their advantages and effectiveness in carrying different drugs, pointing out relevant issues. The authors also describe the challenges in overcoming the lungs barriers and how a proper LNP design could help. Moreover, they list the LNPs currently approved in clinical trial also reporting them in a table. In conclusion this paper is a nice addition to the topic and worth of publication on this journal. It does have only one figure and one table and adding more may make it more appealing. The following are only few suggestions, a couple, all or different ones could be included:
· a table of the successful therapeutics
· a table or schematic of the challenges and/or advantages of LNPs for inhaled therapeutics
· a table of the drugs successfully encapsulated in inhaled LNP therapeutics
· a table of LNP design choices and/or formulations
Please, also make sure to spell out CF the first time it is used instead than at the following paragraph at the end of page 5.
Author Response
Point-to-point response to reviewers’ comments (in blue):
Reviewer 1
Comments and Suggestions for Authors
The review entitled “Lipid Nanoparticles as Delivery Vehicles for Inhaled Therapeutics” is well written and nicely summarize the use of lipid nanoparticles (LNPs) for inhaled therapeutics highlighting the successful attempts and limitations, referring to an extensive past and recent literature. It focuses the application on few most deadly lung diseases, describing the use of different formulations and structures of LNPs and their advantages and effectiveness in carrying different drugs, pointing out relevant issues. The authors also describe the challenges in overcoming the lungs barriers and how a proper LNP design could help. Moreover, they list the LNPs currently approved in clinical trial also reporting them in a table. In conclusion this paper is a nice addition to the topic and worth of publication on this journal. It does have only one figure and one table and adding more may make it more appealing. The following are only few suggestions, a couple, all or different ones could be included:
- a table of the successful therapeutics
- a table or schematic of the challenges and/or advantages of LNPs for inhaled therapeutics
- a table of the drugs successfully encapsulated in inhaled LNP therapeutics
- a table of LNP design choices and/or formulations
Response:
We thank the reviewer for this suggestion. The successful therapeutics approved or have gone through Phase III clinical trials have already been presented in the original Table 1.
As discussed in our review, using LNP for inhaled therapeutics development is still in relatively early stages, so the LNP design choices and/or formulations used for inhaled delivery are still limited and require further studies. Hence, we think tables in these areas are not feasible.
We have added a table on the advantages and challenged of LNPs for inhaled therapeutics as Table 1. The original Table 1 is now Table 2 in the revised manuscript.
Please, also make sure to spell out CF the first time it is used instead than at the following paragraph at the end of page 5.
Response: This has been corrected this accordingly.
Reviewer 2 Report
Here I present the review of the paper entitled “Lipid Nanoparticles as Delivery Vehicles for Inhaled Therapeutics” submitted to Biomedicines.
This is well written review regarding usage of LNPs in lung diseases.
In my opinion novelty of the paper is questionable, as several review articles about different biomedical aspects of LNPs have been published recently. Nevertheless, paper may be beneficial for the readers. Thus I can support it publication after minor revision.
1) (p. 1) “The top five lung diseases include tuberculosis, respiratory infections, lung cancer, asthma and chronic obstructive pulmonary disease (COPD), which together have a huge global burden”. What do you mean by “top five”. Please specify.
2) (p. 2) “(…) severe breathing difficulties and lasting pulmonary scarring [6, John Hopkins 7, 8, 9].” Please correct citation in mentioned and following sentence.
3) (p. 4) “Doxil® achieved accumulation within tumours, greater anti-tumour efficacy and reduced cardiotoxicity characteristic of free doxorubicin due to the enhanced permeability and retention effect”. Please describe EPR effect in more depth.
4) In part about lung diseases CF, LAM, LAH are not described, but they appear later in the text.
Author Response
This is well written review regarding usage of LNPs in lung diseases.
In my opinion novelty of the paper is questionable, as several review articles about different biomedical aspects of LNPs have been published recently. Nevertheless, paper may be beneficial for the readers. Thus I can support it publication after minor revision.
1) (p. 1) “The top five lung diseases include tuberculosis, respiratory infections, lung cancer, asthma and chronic obstructive pulmonary disease (COPD), which together have a huge global burden”. What do you mean by “top five”. Please specify.
Response: We have clarified this in the revised manuscript.
2) (p. 2) “(…) severe breathing difficulties and lasting pulmonary scarring [6, John Hopkins 7, 8, 9].” Please correct citation in mentioned and following sentence.
Response: We have corrected this.
3) (p. 4) “Doxil® achieved accumulation within tumours, greater anti-tumour efficacy and reduced cardiotoxicity characteristic of free doxorubicin due to the enhanced permeability and retention effect”. Please describe EPR effect in more depth.
Response: We have added the explanation of EPR in the text.
4) In part about lung diseases CF, LAM, LAH are not described, but they appear later in the text.
Response: All have been rectified.
Reviewer 3 Report
This review article present a valuable discussion of the potential of using LNP as a pulmonary drug delivery method. The article shows the newest information. In my opinion, this article can be accepted for publication in Biomedicines in the presence form.
Author Response
No revision needed from this reviewer.